# Thermogenic Characterization and Antifungal Susceptibility of *Candida auris* by Microcalorimetry

**DOI:** 10.3390/jof5040103

**Published:** 2019-11-06

**Authors:** Mariagrazia Di Luca, Anna Koliszak, Svetlana Karbysheva, Anuradha Chowdhary, Jacques F. Meis, Andrej Trampuz

**Affiliations:** 1BIH Center for Regenerative Therapies, Charité—Universitätsmedizin Berlin, 10117 Berlin, Germany; mariagrazia.diluca@unipi.it (M.D.L.); anna.koliszak@charite.de (A.K.); svetlana.karbysheva@charite.de (S.K.); 2Department of Biology, University of Pisa, via San Zeno 37, 56127 Pisa, Italy; 3Charité—Universitätsmedizin Berlin, corporate member of Freie Universität Berlin, Humboldt-Universität zu Berlin and Berlin Institute of Health, Center for Musculoskeletal Surgery, 10117 Berlin, Germany; 4Department of Medical Mycology, Vallabhbhai Patel Chest Institute, University of Delhi, Delhi 11007, India; chowdhary.anuradha@gmail.com; 5Department of Medical Microbiology and Infectious Diseases, Canisius-Wilhelmina Hospital, 6532 SZ Nijmegen, The Netherlands; 6Centre of Expertise in Mycology Radboudumc/Canisius-Wilhelmina Hospital, 6532 SZ Nijmegen, The Netherlands

**Keywords:** *Candida auris*, biofilm, antifungal resistance, isothermal microcalorimetry, Metschnikowiaceae clade

## Abstract

*Candida auris* has emerged globally as a multidrug-resistant fungal pathogen. Isolates of *C. auris* are reported to be misidentified as *Candida haemulonii*. The aim of the study was to compare the heat production profiles of *C. auris* strains and other *Candida* spp. and evaluate their antifungal susceptibility using isothermal microcalorimetry. The minimum heat inhibitory concentrations (MHIC) and the minimum biofilm fungicidal concentration (MBFC) were defined as the lowest antimicrobial concentration leading to the lack of heat flow production after 24 h for planktonic cells and 48 h for biofilm-embedded cells. *C. auris* exhibited a peculiar heat production profile. Thermogenic parameters of *C. auris* suggested a slower growth rate compared to *Candida lusitaniae* and a different distinct heat profile compared to that of *C. haemulonii* species complex strains, although they all belong to the Metschnikowiaceae clade. Amphotericin B MHIC and MBFC were 0.5 µg/mL and ≥8 µg/mL, respectively. *C. auris* strains were non-susceptible to fluconazole at tested concentrations (MHIC > 128 µg/mL, MBFC > 256 µg/mL). The heat curve represents a fingerprint of *C. auris*, which distinguished it from other species. Treatment based on amphotericin B represents a potential therapeutic option for *C. auris* infection.

## 1. Introduction

Since its initial description in 2009, *Candida auris* has emerged as a new nosocomial pathogen, characterized by traits of drug resistance, that may cause fungemia and other deep-seated infections in at-risk populations, posing a global threat for public health [1,2]. Genetic analyses have placed *C. auris* within the Metschnikowiaceae family, most closely related to species of the *Candida haemulonii* complex and *Candida lusitaniae*, with a marked divergence from other *Candida* species. Unlike most other *Candida* spp., *C. auris* is associated with skin rather than with gastrointestinal tract colonization and presents an intrinsic resistance to conventional front-line antifungal agents, antiseptics, and disinfectants [1]. In fact, resistance to azole is rather common among circulating strains of *C. auris,* at least in South Asia, and clones resistant to amphotericin B and echinocandins have also been described [1,3,4]. *C. auris* is able to survive and persist in different environments and conditions with a high grade of resilience and adaptivity [5]. Such a persistent colonizing phenotype of *C. auris* may also be linked to its ability to form a biofilm [6,7,8]. Analogously to other *Candida* spp., *C. auris* has been shown to attach on abiotic surfaces and form biofilms in vitro [8]. In vitro biofilms of *C. auris* appeared less thick than those of *Candida albicans*, mostly due to the lack of hyphae production [9]. Recently, the involvement of *C. auris* in an intra-articular infection (usually due to biofilm formation) of a woman with a long-term ankle spacer has also been reported [10].

Due to the ability of *C. auris* to spread among patients, colonize the hospital environment rapidly, and easily develop resistance to antifungals, an accurate and timely identification of species paired to an appropriate determination of *C. auris* antifungal susceptibilities is the key to control outbreaks and antifungal therapy [11]. Although *C. auris* was discovered 10 years ago [12], it is thought to have been misidentified as *C. haemulonii* on several occasions by commercial identification systems [13], suggesting that *C. auris* has likely been circulating as a human pathogen before 2009. Indeed, the failure in the misidentification of *C. auris* is mainly due to systems based on biochemical methods (such as VITEK 2, Phoenix, and API20C AUX). Matrix-assisted laser desorption ionization–time of flight mass spectrometry (MALDI-TOF MS) is a specialized proteomic-based technique, which represents a rapid, accurate, and convenient technology for the identification of microorganisms at the species level. MALDI-TOF MS based on the comparison of the generated spectra for each sample with the reference database can differentiate *C. auris* from other *Candida* species if the database is fully updated with the information of that species [10]; otherwise no identification or misidentification will be obtained as shown recently with external quality assessment [13,14,15].

Isothermal microcalorimetry (IMC) is a nonspecific analytical tool for the measurement of heat produced or consumed by chemical reactions or physical changes of state in a specimen ampoule, including heat generated by complex biological processes in cultured microbial cells [16]. If microorganisms replicate in a microcalorimetry ampoule and produce enough metabolic heat to exceed the instrument’s detection limit, their exponential growth phase can be detected and observed in real time. Each microorganism produces a peculiar thermogenic profile which represents a sort of fingerprinting of the species or strains [17]. Moreover, IMC has been widely employed to evaluate the metabolic heat produced by both bacteria and fungi and to test their ability to form biofilms on different materials [18,19]. IMC analysis also proved helpful to evaluate the activity of different antimicrobials, including antibiotics [20,21,22], bacteriophages [23,24], and antifungals [25] against planktonic and sessile microbial cells. To the best of our knowledge, the thermogenic characteristics and the antifungal susceptibilities of planktonic and biofilm *C. auris* have not been investigated by microcalorimetry, yet.

Therefore, the aim of the study was to analyze the metabolic heat profiles produced by different clinical strains of *C. auris* in comparison to other *Candida* spp., including those belonging to the same Metschnikowiaceae family, and evaluate their susceptibility to amphotericin B and to fluconazole in planktonic and biofilm forms by IMC.

## 2. Materials and Methods

### 2.1. Candida Strains and Growth Conditions

*C. albicans* ATCC 90028, *Candida glabrata* DSY 562, *Candida parapsilosis* ATCC 22019 and clinical strains of *C. haemulonii* (H 433), *Candida duobushaemulonii* (D-437), *C. lusitaniae* (L-719), *Candida pseudohaemulonii* (P-430), *Candida kefyr* (K-629), *Candida tropicalis* (T 317), and *C. auris* (reference strain CBS14916 and clinical strains 10051257, 10051259, 10051266, 10051297) were use in this study (Table 1). Based on whole-genome sequencing, five different clades of *C. auris* have been described by region (East Asian, South Asian, African, South American, and Iranian) [26]. Yeasts were stored using a cryovial bead preservation system (Microbank; Pro-Lab Diagnostics, Richmond Hill, ON, Canada) at −80 °C. To find out the best growth conditions, all strains were incubated on Sabouraud dextrose agar (Oxoid) at 25, 30, and 37 °C for 24, 48, and 72 h. After each incubation time point, the colony size was manually measured by a ruler and a picture was taken. For all antifungal tests, five *C. auris* strains were grown on Sabouraud dextrose agar at 37 °C for 48 h and the used liquid medium was RPMI 1640 (Thermo Fisher Scientific, Waltham, MA, USA). The standard inoculum for all tests was ≈5 × 10^5^ CFU/mL.

### 2.2. Antimicrobial Agents

The powders of sodium-deoxycholate amphotericin B (Sigma-Aldrich Chemie, Taufkirchen, Germany) and fluconazole (Pfizer, Freiburg, Germany) were used in this study. Amphotericin B and fluconazole were dissolved in a 2 g/L stock solution with distilled water and stored at −20 °C until use.

### 2.3. Biofilm Formation on Porous Glass Beads

The biofilm forming ability of *C. auris* was evaluated according to a protocol previously described for other *Candida* species, including *C. albicans* [27]. Briefly, *C. auris* CBS14916 was propagated as a biofilm on porous glass beads with a diameter of 2–4 mm, porosity of 0.2 m^2^/g, and pore size of 60 μm (VitraPor; ROBU, Hattert, Germany) by inoculating two to three colonies of the yeast either in Sabouraud dextrose broth or RPMI 1640 medium (one bead per 1 mL medium) and incubating beads at 37 °C for 24, 48, or 72 h, respectively. After each time point, beads were washed three times with 0.9% saline solution to remove all planktonic cells and sonicated to dislodge biofilm-embedded yeasts, as previously described [25,28]. Then, sonication fluids were plated onto Sabouraud dextrose agar which were incubated for 48 h at 37 °C for colony counting [25]. *C. albicans* biofilms were prepared with the same conditions and were used as control.

### 2.4. Characterization and Antifungal Assay by Isothermal Microcalorimetry (IMC)

The metabolically related heat produced by the different *Candida* species was measured using an isothermal 48-channel batch calorimeter (TAM III; TA Instruments, Newcastle, DE, USA), as described previously [27]. Yeast cells of each species (5 × 10^5^ CFU/mL final inoculum) were inoculated in Sabouraud dextrose broth and incubated in sterile ampoules, which were then placed in the calorimeter. The heat produced was monitored at 37 °C for 48 h.

To determine antifungal activity against planktonic *C. auris*, yeast cells (5 × 10^5^ CFU/mL final inoculum) were inoculated in microcalorimetry ampoules filled with RPMI 1640 and twofold serial dilutions of amphotericin B (0.125–2 µg/mL) and fluconazole (16–128 µg/mL). Heat production was measured at 37 °C for 48 h. The minimum heat inhibiting concentration (MHIC) for planktonic yeasts was defined as the lowest antifungal concentration that inhibited the yeast growth-related heat production during 24–48 h incubation in the microcalorimeter.

To determine cidal activity of the antifungal agents against planktonic yeasts, *C. auris* suspensions treated with drugs were plated onto Sabouraud dextrose agar and incubated for 48 h at 37 °C, after microcalorimetry measurements. The minimum fungicidal concentration (MFC) was defined as the lowest antimicrobial concentration that showed a reduction ≥3 log_10_ in CFU/mL, as compared to the CFU/mL number of the initial inoculum.

For the determination of antifungal activity against *C. auris* biofilm, biofilm beads, obtained as previously described, were exposed to twofold dilutions of either amphotericin B (1–16 µg/mL) or fluconazole (128 and 256 µg/mL) in RPMI 1640 medium and incubated again for 24 h at 37 °C. After treatment, the glass beads were rinsed three times with 0.9% saline and transferred into microcalorimetry ampoules with 3 mL fresh RPMI 1640 medium to measure the re-growth of surviving cells of biofilms still attached to the bead. Beads without yeasts were used as sterility control. The heat production was measured at 37 °C for 48 h. The minimum biofilm fungicidal concentration (MBFC) was defined as the lowest antifungal concentration that strongly reduced the number of viable yeast cells within the biofilm, therefore leading to undetectable heat flow values. The minimum biofilm eradicating concentration (MBEC) was defined as the lowest antifungal concentration that eradicates all sessile biofilm cells (0 CFU/bead on plate counts), evaluated by CFU counting after sonication of beads, as previously described [16,28]. All experiments were performed in triplicate.

### 2.5. Data Analysis

The resulting data were expressed as heat flow (μW) versus time (h) and as heat (J) versus time (h). Figures were plotted using GraphPad Prism 8.0 (GraphPad Software, La Jolla, CA, USA). IMC time to detection (TTD, h) was defined as the time between the insertion of the ampoule into the microcalorimeter and the exponentially increasing heat flow production exceeding the threshold of 10 μW [27].The maximum heat flow peak (P_max_, μW), the time of the maximum heat flow peak (T_max_, h), and the total heat produced (H_tot_, J) were defined as the highest value of the heat flow–time curve, the time at which the P_max_ was detected, and the cumulative amount of heat produced during the whole experiment, respectively.

IMC data were converted into microbiologically relevant information such as growth rate (µ, J/h) and lag phase (λ, h) by deriving according to growth models, as previously reported [17,22,29,30].

## 3. Results

### 3.1. Microcalorimetric Analysis of Candida spp.

The metabolic heat produced by planktonic *Candida* strains during their growth was monitored in real time for 48 h at 37 °C by isothermal microcalorimetry. Figure 1 shows representative thermogenic profiles of heat flow (Figure 1a) and total heat, (Figure 1b) obtained by different *Candida* species including *C. auris* and other strains in the Metschnikowiaceae clade. In Table 2, values of different thermogenic parameters (related to curves of Figure 1) are reported. *C. parapsilosis* ATCC 22019, *C. glabrata* DSY 562, and *C. kefyr* (K-629) showed a similar profile of the heat production curve, although they exhibited a different P_max_ (195.34, 168.44, and 145.10 µW, respectively). All the three species also were characterized by a similar T_max_, ranging from 6.5 to 7 h. *C. lusitaniae* (L-719), one of the Metschnikowiaceae clade, displayed a comparable curve to *C. albicans* (ATCC 90028) with a similar P_max_, but with an early T_max_ (∆T_max_ = 4.19 h). In comparison to the above-mentioned *Candida* species, *C. auris* (CBS14916) and *C. tropicalis* (T-317) showed a lower P_max_ and a delayed T_max_. However, in most *Candida* species, the total heat produced was ranging between 4.71 and 5.60 J, except in *C. tropicalis*, where the total heat during 48 h was 8.49 J. Among *C. auris* clinical strains, all four isolates showed similar heat flow and total heat curves as shown in Figure 2. A different behavior in the heat production has been observed in samples containing *C. duobushaemulonii* (D-437), *C. haemulonii* (H-433), and *C. pseudohaemulonii* (P-430) at tested experimental conditions. All showed a P_max_ lower than all other *Candida* species analyzed (23.86–29.48 µW) at the last measured time point (46 h); thus, their total heat values were also lower. Even the lag phase for these three *Candida* species was similar, ranging between 17 and 22 h which was in contrast with the lag phase values of other strains ranging between 3.96 and 5.69 h. For most of the strains, the time to detection (TTD) of the heat produced was ranging from 1.01 to 2.5 h. Only the TTD of *C. haemulonii* was delayed (7.21 h). *C. auris*, *C. pseudohaemulonii*, and the strains of the *C. haemulonii* complex exhibited a slower growth rate (0.23–0.14 J/h) at 37 °C in comparison to all the other *Candida* species (1.04–0.38 J/h).

### 3.2. Analysis of Candida spp. Growth on Solid Medium

The ability of *C. auris*, *C. duobushaemulonii, C. haemulonii*, and *C. pseudohaemulonii* to grow on agar medium at 37 °C was also impaired as suggest by the small size of *C. auris* colonies (CBS14916), manually measured, and the absence of any colonies of *C. pseudohaemulonii*, *C. haemulonii*, and *C. duobushaemulonii* after 24 h of plate incubation (Figure 3 and Table 3).

### 3.3. Analysis of C. auris Biofilm Formation

The ability of *C. auris* (CBS14916) to form biofilm was evaluated as CFU number of yeast cells detached by washed porous glass beads (to remove planktonic cells) after 24, 48, and 72 h incubation of *Candida* cells either in Sabouraud or in RPMI 1640 dextrose broth at 37 °C (Figure 4). *C. albicans* ATCC 90028 was used as a control of biofilm formation on the same material. After 24 h incubation, the number of *C. auris* CFU ranged between 10^5^ and 10^6^ CFU/mL in both media which was comparable to the number of *C. albicans* yeasts detached, suggesting that no difference in biofilm cell growth occurred in Sabouraud and RPMI 1640. The number of *Candida* cells removed from beads remained similar even in the case of a prolonged time of biofilm formation. This has been observed for both investigated *Candida* species.

### 3.4. Susceptibility Assay of Planktonic and Biofilm-embedded C. auris to Amphotericin B and Fluconazole by IMC

Antifungal activity of amphotericin B and fluconazole was assayed against planktonic (Figure 5) and biofilm (Figure 6) yeasts of *C. auris* by IMC. Values of MHIC, MFC, MBFC, and MBEC are reported in Table 4. All planktonic strains of *C. auris* were susceptible to 0.5 µg/mL amphotericin B with MFC ranging between 0.5 and 1 µg/mL. While, MBFC values for amphotericin B were higher than MHIC, resulting in 8 µg/mL for *C. auris* 10051257 and 10051297 and >8 µg/mL for *C. auris* 10051259 and 10051266. MBFC values also corresponded to eradicating concentrations (MBEC). By contrast, both planktonic and sessile strains of *C. auris* were non-susceptible to the tested concentrations of fluconazole (128 and 256 µg/mL, respectively).

## 4. Discussion

Among different *Candida* spp., *C. albicans* is the main etiological agent of candidiasis [31]. However, non-*albicans Candida* species have recently gained scientific and epidemiological interest as their frequency of detection is increasing worldwide. This shift to non-*albicans Candida* species has also increased with the increasing number of diagnosis of infection due to *C. auris*. *C. auris* is a newly emerging multidrug-resistant fungal pathogen, associated with severe invasive infection and outbreaks [5,32,33,34,35]. As *C. auris* is associated to high mortality rates, an accurate and prompt identification of this species and the susceptibility/resistance evaluation is fundamental to set the appropriate antifungal therapy [13]. Here, we used IMC to analyze the metabolic heat produced by *C. auris* and compare it to that produced by *C. albicans* and non-*albicans Candida* species. Our data suggest that at 37 °C in our experimental conditions, *C. auris* is able to replicate slower (µ = 0.23 J/h; generation time 180 min) than other *Candida* species (*C. albicans* µ = 0.38 J/h; generation time = 108 min), but faster than *C. pseudohaemulonii* (µ = 0.14 J/h; generation time = 297 min) and *C. haemulonii* (µ = 0.16 J/h; generation time = 259 min). Moreover, the ability of *C. auris* to grow on solid medium and form visible colonies at different temperatures and within 24–48 h suggests that this species is more adapted to high temperature in comparison to other species belonging to the same family. This observation is in agreement with the analysis of phylogenetic and thermotolerance performed on *C. auris*, where the increase in ambient temperatures as a result of global warming may have acclimatized the organism to adapt to and survive at different host temperatures [36,37]. *C. auris* showed a peculiar thermogenic profile, with a lower and larger heat flow curve compared to that observed in other *Candida* species. Within the same species, all clinical strains of *C. auris* showed a similar thermogenic profile. Although five is a low number, the analysis of additional *C. auris* isolates might confirm the capability of IMC to discriminate *C. auris* by its characteristic metabolic heat curve, which represents a unique fingerprinting for this species. A limitation of our study is that we only included isolates belonging to the largest Clade I. Potential phenotypic variation between different clades may show other thermogenic profiles.

As reported by different authors, *C. auris* is able to form biofilms in vitro and probably in vivo [7,8,9]. Sherry and colleagues first identified that *C. auris* was able to produce intermediate quantities of biomass compared to *C. albicans* (high producer) and *C. glabrata* (low producer) [9]. We confirm this observation by letting *C. auris* (reference strain) form a biofilm on porous glass beads, a technique widely used by our group to study bacterial [21,22,38,39] and fungal biofilms, including the susceptibility of *C. albicans* to different antifungal agents [25,27], and performing analysis by IMC. To enumerate yeast cells attached on the beads we used sonication to detach yeast cells from the material and disperse yeast aggregates [25]. By using *C. albicans* biofilm as control, we observed that *C. auris* was able to attach on porous glass beads and form a sessile community as suggested by the same CFU number recovered after sonication of both species. Although we would need microscopy to evaluate the presence of extracellular matrix and 3D-structure of a biofilm, in analogy to *C. albicans*, we are confident that *C. auris* also established biofilms on the beads. As the increase in time of yeasts/beads incubation did not change the number of CFUs recovered, we used this incubation for antifungal susceptibility experiments. As *C. auris* was recently discovered, no definitive MIC breakpoints exist, but tentative breakpoints and epidemiological cutoff valueshave been suggested [40,41]. Regarding fluconazole, we observed a correlation between MIC values obtained by VITEK2 and MHIC values observed by IMC, in both cases higher than 32 µg/mL the tentative breakpoint for nonsusceptibility. These values are in agreement with published susceptibility data which suggest that most circulating *C. auris* strains are characterized by high fluconazole MICs (>64 µg/mL) and variable susceptibility to amphotericin B and echinocandins [4,13,41]. By contrast, a discrepancy was seen in MHIC for amphotericin B in comparison to the MIC for four of five strains. MIC values obtained by VITEK 2 for all strains, except *C. auris* 10051259 were 16 times (or more) higher than those of MHIC, thus all these strains of *C. auris* were resistant to amphotericin B, based on the commercial system. The evaluation of the antibiofilm activity of amphotericin B by IMC revealed that all *C. auris* species were susceptible to higher concentrations of the polyene (MBFC and MBEC ≥8 µg/mL) compared to the MHIC obtained for planktonic yeast. This is in agreement with data recently reported in the literature [6]. Despite failure to produce biofilms as robust as those of *C. albicans*, sessile *C. auris* were shown to tolerate higher concentrations of amphotericin B, although its planktonic counterpart was susceptible to lower concentrations of the drug [9]. Moreover, Kean et al. suggested that biofilm-embedded cells were phenotypically tolerant to all classes of antifungals [6]. By transcriptional analysis of *C. auris* genes, they found that tolerance is correlated with increase of efflux pumps gene expression for both ATP-binding cassette and major facilitator superfamily (MFS) transporters and the upregulation of genes encoding for molecules involved in extracellular matrix formation [7]. Although these two phenomena explain resistance to azoles well, it might be that these phenomena also contribute to polyene tolerance. The mechanisms employed by *C. auris* to survive and persist in the environment are unknown, but there is evidence to suggest that biofilm formation may play a key role, as has been suggested for other microorganisms well adapted to hostile conditions in biofilms. In conclusion, IMC was useful to distinguish *C. auris* from other *Candida* species, such as *C. haemulonii* and it might be useful in the future to investigate more in depth the behavior of *C. auris* biofilm on different materials. Treatment based on amphotericin B represents a potential therapeutic option for *C. auris* infection, which can be easily tested by using IMC.

## Figures and Tables

**Figure 1 jof-05-00103-f001:**
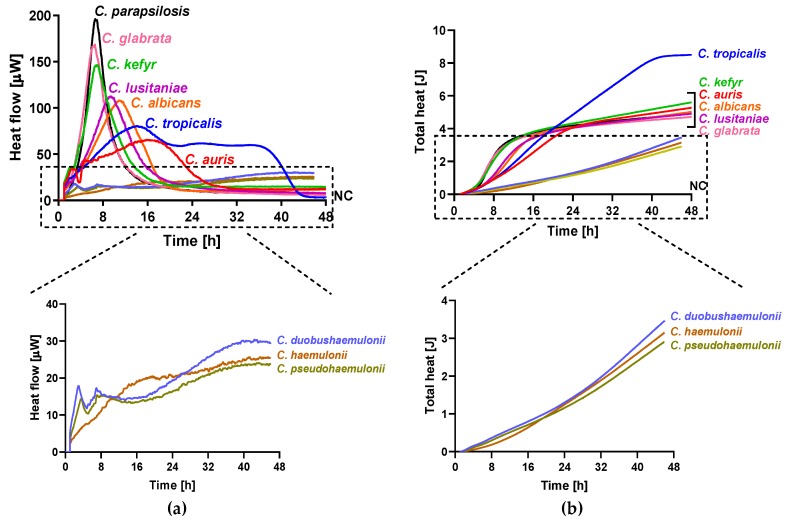
Microcalorimetry analysis of *Candida* spp. Heat flow (**a**) and total heat (**b**) curves generated by planktonic *C. parapsilosis*, *C. glabrata*, *C. kefyr*, *C. lusitaniae*, *C. albicans*, *C. tropicalis*, *C. auris*, *C. duobushaemulonii*, *C. haemulonii*, and *C. pseudohaemulonii* in RPMI 1640 at 37 °C.

**Figure 2 jof-05-00103-f002:**
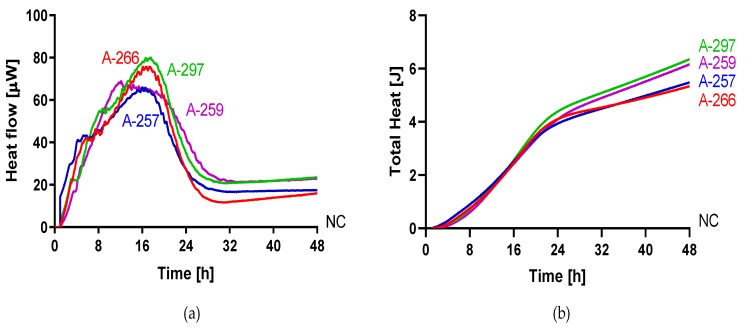
Microcalorimetry analysis of *C. auris*. Heat flow (**a**) and total heat (**b**) curves generated by planktonic *C. auris* 10051257, *C. auris* 10051259, *C. auris* 10051266, and *C. auris* 10051297 in RPMI 1640 at 37 °C.

**Figure 3 jof-05-00103-f003:**
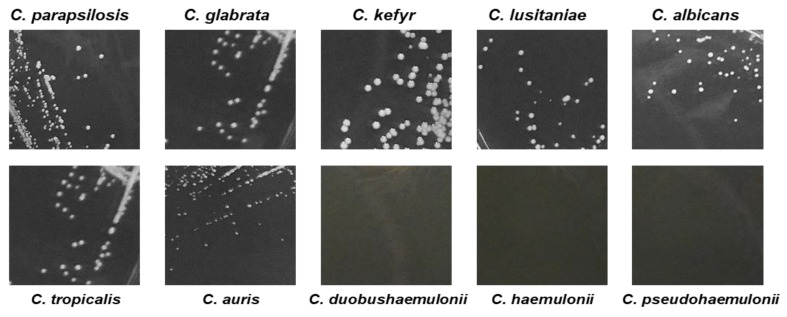
Pictures of *Candida* spp., colonies growth onto Sabouraud dextrose agar. *C. auris* exhibited smaller colony size compared to most *Candida* spp. after 24 h incubation time at 37 °C.

**Figure 4 jof-05-00103-f004:**
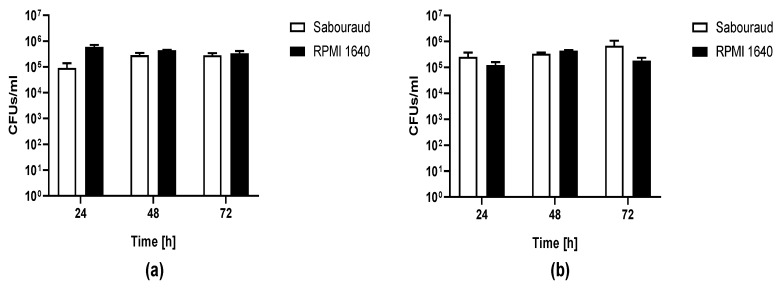
Biofilm of *C. auris* CBS14916 (**a**) and *C. albicans* ATCC 90028 (**b**) formed on porous glass beads. CFU number of *C. auris* and *C. albicans* dislodged from glass beads by sonication after incubation in either Sabouraud or RPMI 1640 broth at 37 °C for 24, 48, and 72 h.

**Figure 5 jof-05-00103-f005:**
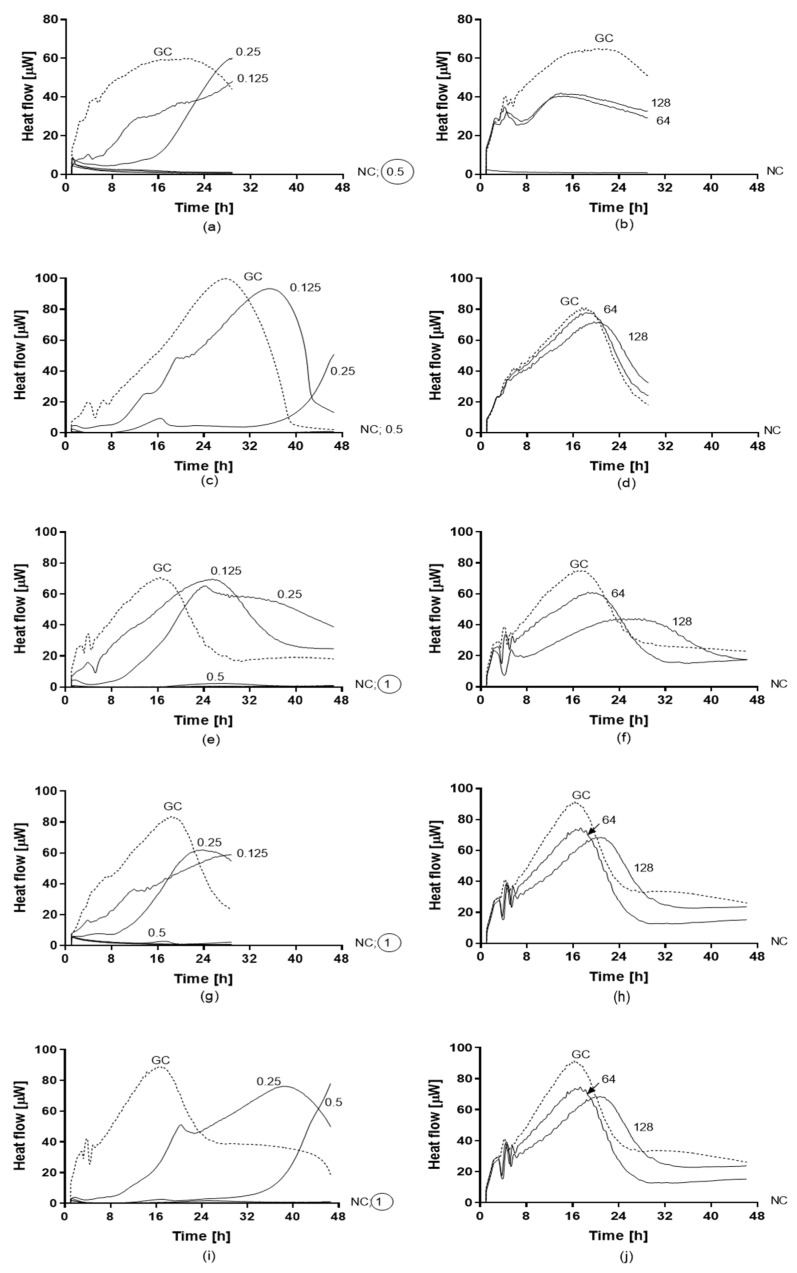
Heat flow curves generated by planktonic strains of *C. auris* CBS14916 (**a**,**b**), *C. auris* 10051257 (**c**,**d**), *C. auris* 10051259 (**e**,**f**), *C. auris* 10051266 (**g**,**h**), and *C. auris* 10051297 (**i**,**j**) in the presence of different concentrations of amphotericin B (**a**,**c**,**e**,**g**) and fluconazole (**b**,**d**,**f**,**h**). Numbers indicate the antifungal concentrations (μg/mL). GC: growth control (without antifungal agent). MHIC is defined as the lowest antifungal concentration that inhibited the yeast growth-related heat production during 24–48 h incubation in the microcalorimeter. The circled value denotes the minimum fungicidal concentration (MFC).

**Figure 6 jof-05-00103-f006:**
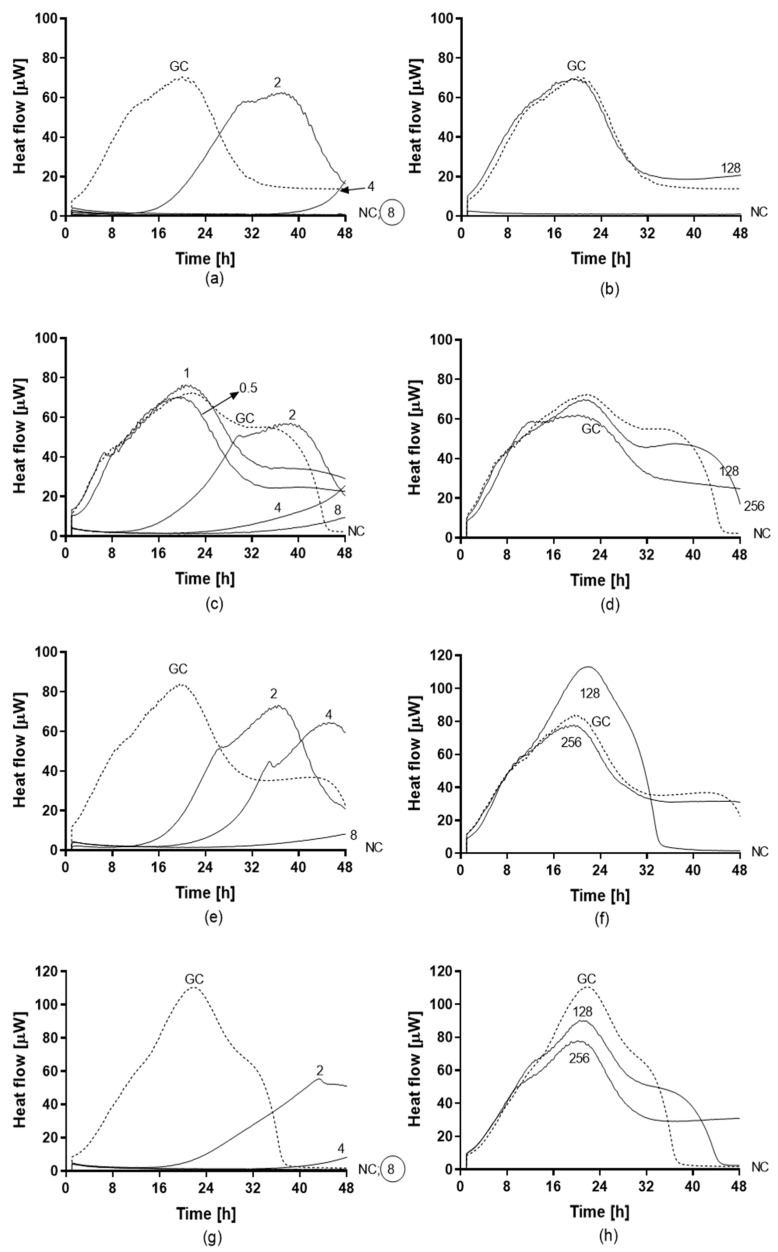
Heat flow curves generated by biofilm-embedded yeasts of *C. auris* 10051257 (**a**,**b**), *C. auris* 10051259 (**c**,**d**), *C. auris* 10051266 (**e**,**f**), and *C. auris* 10051297 (**g**,**h**) after treatment with different concentrations of amphotericin B (**a**,**c**,**e**,**g**) and fluconazole (**b**,**d**,**f**,**h**). Numbers indicate the antifungal concentrations (μg/mL). GC: growth control (without any antifungal treatment). The minimum biofilm fungicidal concentration (MBFC) was defined as the lowest antifungal concentration which led to undetectable heat flow values. The circled value denotes the minimum biofilm eradicating concentration (MBEC).

**Table 1 jof-05-00103-t001:** *Candida* clinical strains used in this study.

Strains	^1^FLC	^2^VRC	^3^AMB	^4^CAS	^5^5FC	^6^MFG	^7^ITC	^8^POS	^9^AFG	^10^ISA
*C. lusitaniae* (L-719)	≤0.5	≤0.12	0.5	0.25	2	0.25	0.012	0.004	0.16	0.004
*C. kefyr* (K-629)	1	≤0.12	1	≤0.12	4	0.12	0.023	0.023	0.12	0.004
*C. tropicalis* (T 317)	1	≤0.12	0.5S	≤0.12	≤1	≤0.06	0.012	0.008	0.008	0.006
*C. haemulonii* (H 433)	2	≤0.12	≥16	0.25	≤1	0.25	n.a.	n.a.	0.5	n.a.
*C. duobushaemulonii* (D-437)	2	≤0.12	2	≤0.12	≤1	≤0.06	n.a.	n.a.	0.032	n.a.
*C. pseudohaemulonii* (P-430)	n.a.	n.a.	n.a.	n.a.	n.a.	n.a.	n.a.	n.a.	n.a.	n.a.
*C. auris* 10051257(467/P/14)	32	2	1.5	≥8	≤1	≥8	1	0.25	≥8	0.5
*C. auris* 10051259(471/P/14)	32	1	1	0.25	≤1	0.125	2	038	0.064	1
*C. auris* 10051266(1113/P/13)	32	1	1	≥8	≤1	≥8	1	0.25	32	0.5
*C. auris* 10051297(550/P/14)	32	2	8	≥8	≥64	≥8	2	0.125	8	0.25
*C. auris* CBS14916	32	0.25	8	0.25	≥64	0.12	0.064	0.032	0.125	0.032

^1^ Fluconazole (FLC); ^2^ voriconazole (VRC); ^3^ amphotericin B (AMB); ^4^ caspofungin (CAS); ^5^ flucytosine (5FC); ^6^ micafungin (MFG); ^7^ itraconazole (ITC); ^8^ posaconazole (POS); ^9^ anidulafungin (AFG); ^10^ isavuconazole (ISA). All *C. auris* originated from A. Chowdhary, Vallabhbhai Patel Chest Institute (VPCI), Delhi, India. Based on whole-genome sequencing, five different clades of *C. auris* have been described by region (East Asian, South Asian, African, South American, and Iranian) [26].

**Table 2 jof-05-00103-t002:** Microcalorimetric parameters of planktonic *Candida* species.

Strains	P_max_ (μW) ^1^	T_max_ (h) ^2^	H_tot_ (J) ^3^	TTD (h) ^4^	λ (h) ^5^	(μ, J/h) ^6^
*C. parapsilosis* ATCC 22019	195.3	6.89	4.86	1.79	5.06	1.04
*C. glabrata* DSY 562	168.4	6.54	4.71	1.05	3.96	0.60
*C. kefyr* (K-629)	145.1	6.54	5.60	1.01	4.00	0.52
*C. lusitaniae* (L-719)	112.3	9.31	4.90	1.01	5.14	0.40
*C. albicans* ATCC 90028	108.1	10.91	5.02	1.01	5.69	0.38
*C. auris* CBS14916	65.1	17.07	5.27	1.01	5.19	0.23
*C. tropicalis* (T 317)	80.3	13.98	8.49	1.01	5.51	0.28
*C. duobushaemulonii* (D-437)	29.5	46.00	3.46	1.69	21.99	0.18
*C. haemulonii* (H 433)	23.9	46.00	3.15	7.21	17.64	0.16
*C. pseudohaemulonii* (P-430)	25.4	46.00	2.91	2.50	20.94	0.14

^1^ P_max_ (μW): the maximum heat flow peak; ^2^ T_max_ (h): time of the maximum heat flow peak; ^3^ H_tot_ (J): total heat produced in J; ^4^ TTD (h): time to detection in h; ^5^ λ (h): lag phase; ^6^ μ (J/h): growth rate.

**Table 3 jof-05-00103-t003:** Average of colony size of *Candida* species grown at different temperatures.

Strains	25 °C			30 °C			37 °C		
	24 h	48 h	72 h	24 h	48 h	72 h	24 h	48 h	72 h
*C. parapsilosis* ATCC 22019	n.g ^1^.	1	2	0.5	1–2	3	1	2	3
*C. glabrata* DSY 562	n.g.	1	2	0.5	2	3	1	2	3
*C. kefyr* (K-629)	n.g.	1	2	0.5	1	1–2	1	2–3	3
*C. lusitaniae* (L-719)	n.g.	0.5	1	0.5	1	1	0.5	1–2	2
*C. albicans* ATCC 90028	n.g.	0.5	1	0.5	1–2	2	0.5	1–2	2
*C. auris* CBS14916	n.g.	<0.5	1	<0.5	1	1	0.5	1–2	3
*C. tropicalis* (T 317)	n.g.	1	2	0.5	1	1	1	1–2	2
*C. duobushaemulonii* (D-437)	n.g.	<0.5	1	n.g.	n.g.	n.g.	n.g.	0.5	1
*C. haemulonii* (H 433)	n.g.	0.5	1	0.5	0.5	1	n.g.	<0.5	0.5
*C. pseudohaemulonii* (P-430)	n.g.	0.5	1	<0.5	0.5	0.5	n.g	<0.5	1

^1^ n.g.: no growth. Numbers referring to colony size in mm.

**Table 4 jof-05-00103-t004:** Values of MHIC, MBFC, MFC, and MBEC (μg/mL) of amphotericin B and fluconazole against planktonic and biofilm *C. auris* strains.

Strains	IMC ^1^				Plating			
	Planktonic		Biofilm		Planktonic		Biofilm	
	MHIC ^2^		MBFC ^3^		MFC ^4^		MBEC ^5^	
	AMB ^6^	FLC^7^	AMB	FLC	AMB	FLC	AMB	FLC
*C. auris 10051257*	0.5	>128	8	>256	1	>128	8	>256
*C. auris 10051259*	0.5	>128	>8	>256	1	>128	>8	>256
*C. auris 10051266*	0.5	>128	>8	>256	1	>128	>8	>256
*C. auris 10051297*	0.5	>128	8	>256	1	>128	8	>256
*C. auris CBS14916*	0.5	>128	n.a.^8^	>256	0.5	>128	n.a.	>256

^1^ IMC: isothermal microcalorimetry; ^2^ MHIC: minimum heat inhibition concentration; ^3^ MBFC: minimum biofilm fungicidal concentration; ^4^ MFC: minimum fungicidal concentration; ^5^ MBEC: minimal biofilm eradication concentration; ^6^ AMB: amphotericin B; ^7^ FLC: fluconazole; ^8^ n.a.: not assayed. Concentration values are expressed as µg/mL.

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
