# Peer review of "Thermogenic Characterization and Antifungal Susceptibility of Candida auris by Microcalorimetry"

_jof, 2019, doi:10.3390/jof5040103_

Round 1

Reviewer 1 Report

Authors aimed to investigate the metabolic heat production profiles of different clinical Candida auris strains and to compare them with other Candida species. Furthermore, antifungal susceptibility profiles of the C. auris strains were measured using isothermal microcalorimetry.

Authors showed that strains of C. auris had a slower growth rate compared to other members of Candida haemuloniae complex. Moreover, a unique thermogenic profile with a lower and larger heat flow curve found in C. auris strains tested in this study.

Manuscript is well-written. It has a clear introduction and detailed methodology. Tables and figures are informative and well-presented. The study has a well-thought design and is indeed based on a novel idea. Results are clear and of interest to a wide readership of the Journal of Fungi.

My minor suggestions and questions to the authors are listed below:

Line 23 in abstract: I suggest using minimum biofilm fungicidal concentration (MBFC). Line 203 in results section: Can authors please clarify the method they used for the measurement of growth on agar medium. Was this done using image analysis software or done manually? As shown in Figure 3, there is a clear variation in colonial size in C. lusitaniae and even C. auris. Are the measurements provided in Table 3 averaged for several measured colonies at each temperature? Line 215 in results section: Can authors please clarify if there was any difference in C. auris biofilm production in Sabouraud or RPMI 1640? Line 277 to 282 in discussion: Can authors please clarify the clade of C. auris isolates used in this study? Are they all from the South Asian clade? If so, do authors think testing strains from other clades may demonstrate different thermogenic profiles? It will be useful if authors comment on this potential phenotypic variation between members of different clades in the discussion. Line 278 in discussion: From evolutionary point of view, can this lower and larger heat flow observed in tested C. auris strains explain any natural adaptation process to infect human host?

Author Response

Reviewer 1

Authors aimed to investigate the metabolic heat production profiles of different clinical Candida auris strains and to compare them with other Candida species. Furthermore, antifungal susceptibility profiles of the C. auris strains were measured using isothermal microcalorimetry.

Authors showed that strains of C. auris had a slower growth rate compared to other members of Candida haemuloniae complex. Moreover, a unique thermogenic profile with a lower and larger heat flow curve found in C. auris strains tested in this study.

Manuscript is well-written. It has a clear introduction and detailed methodology. Tables and figures are informative and well-presented. The study has a well-thought design and is indeed based on a novel idea. Results are clear and of interest to a wide readership of the Journal of Fungi.

RE: Authors thank Reviewer 1 for his/her appreciation of our manuscript and for all suggestions provided.

My minor suggestions and questions to the authors are listed below:

Line 23 in abstract: I suggest using minimum biofilm fungicidal concentration (MBFC).

RE: The sentence has been changed accordingly.

Line 203 in results section: Can authors please clarify the method they used for the measurement of growth on agar medium. Was this done using image analysis software or done manually?

RE: Authors measured colony size manually, by ruler. This information was now included in the Materials & Methods. Now, it reads: “After each incubation time point the colony size were manually measured by a ruler and a picture was taken” and it was also specified in Results section.

As shown in Figure 3, there is a clear variation in colonial size in C. lusitaniae and even C. auris. Are the measurements provided in Table 3 averaged for several measured colonies at each temperature?

RE: The table provided the averaged of colony size, for this reason the title of table 3 ha been changed. Now it reads “Table 3. Average of colony size of Candida species grown at different temperature”

Line 215 in results section: Can authors please clarify if there was any difference in C. auris biofilm production in Sabouraud or RPMI 1640?

RE: Authors added the following sentence in line 220 “suggesting that no difference in biofilm cell growth occurred in Sabouraud and RPMI 1640”

Line 277 to 282 in discussion: Can authors please clarify the clade of C. auris isolates used in this study? Are they all from the South Asian clade? If so, do authors think testing strains from other clades may demonstrate different thermogenic profiles? It will be useful if authors comment on this potential phenotypic variation between members of different clades in the discussion.

RE: All isolates belonged to the South Asian clade I (this information added on line 90-92 and legends table 1, line 164). We included the following statement regarding the limitation of our study by not having included all available clades.

“Based on whole genome sequencing, five different clades of C. auris have been described by region (East Asian, South Asian, African, South American, and Iranian) [Chow, N.A.; de Groot, T.; Badali, H.; Abastabar, M.; Chiller, T.M.; Meis, J.F. Potential fifth clade of Candida auris, Iran, 2018. Emerg Infect Dis. 2019, 25, 1780-1781. doi: 10.3201/eid2509.190686].”

“A limitation of our study is that we only included isolates belonging to the largest Clade I. Potential phenotypic variation between different clades may show other thermogenic profiles.”

Line 278 in discussion: From evolutionary point of view, can this lower and larger heat flow observed in tested C. auris strains explain any natural adaptation process to infect human host?

RE: Indeed, this may explain the differences in C. auris. However, it would be too speculative at this early point to correlate the heat production in a synthetic medium with natural adaptation to infect human. Nevertheless, we added a sentence on this topic in the Discussion.

Reviewer 2 Report

The paper deals with the growth and antimicrobial susceptibility of C.auris. The study uses conventional methods and isothermal microcalorimetry. The work is carefully done, leading to clear results and sound discussion. There are only a few minor points that need to be addressed (see below). Overall, the publication is of good quality and certainly deserves publication.

Minor points

1) Table 2: As experiment were performed in triplicates, standard deviation should be give for the parameters determined. Also for Pmax, the instrument short term noise being of 0.2uW, the accuracy given by the authors is impossible to reach. Therefore I recommend rounding the values 1 digit (I.e, 0.1uW).

2) About colony size, table 3. It seems to me that the authors used very old fashioned methods to determine colony sizes probably ruler or caliper measurements (the MM section is unclear on that point). Although the approach of the authors is correct, it would be nice if the authors would recognize that much better accuracy (and larger sample size) can be obtained using image analysis softwares such as ImageJ. It might have been done in the field of calorimetry already (look at Raivo Vilu’s work or Thomas Maskow’s recent work).

Please expand “After each incubation time point the colony size were measured and a picture was taken”… or use the picture for image analysis.

3) Antimicrobial determination by IMC. Again the work is well done. However for comparison purposes it would be valuable if the authors could provide the same parameters as those used in table 3.

4) Line 270-272: it would be worth using the growth rate and calculating doubling time here. This would certainly help readers to picture the difference in replication speed.

5) Line 322-323: I woud remove the statement on combination therapies as it is speculative at this point and no combinations have been used in this study.

Author Response

Reviewer 2

The paper deals with the growth and antimicrobial susceptibility of C. auris. The study uses conventional methods and isothermal microcalorimetry. The work is carefully done, leading to clear results and sound discussion. There are only a few minor points that need to be addressed (see below). Overall, the publication is of good quality and certainly deserves publication.

Minor points

1) Table 2: As experiment were performed in triplicates, standard deviation should be give for the parameters determined. Also for Pmax, the instrument short term noise being of 0.2uW, the accuracy given by the authors is impossible to reach. Therefore I recommend rounding the values 1 digit (I.e, 0.1uW).

RE: In the Table 2 the thermogenic parameters do not represent the average of the three experiments but related to the representative curves of Figure 1. Therefore, SD were not indicated. Authors specified this point in the main text (in line 170).

Authors agreed with Reviewer 2, the values of Pmax were rounded one digit after the decimal point.

2) About colony size, table 3. It seems to me that the authors used very old fashioned methods to determine colony sizes probably ruler or caliper measurements (the MM section is unclear on that point). Although the approach of the authors is correct, it would be nice if the authors would recognize that much better accuracy (and larger sample size) can be obtained using image analysis softwares such as ImageJ. It might have been done in the field of calorimetry already (look at Raivo Vilu’s work or Thomas Maskow’s recent work). Please expand “After each incubation time point the colony size were measured and a picture was taken”… or use the picture for image analysis.

RE: Authors thank the reviewer for the suggestion. They agree that a software ImageJ could provide a more accurate analysis, however the authors measured colony size manually, by ruler. This information was included in MM and Results. Now, in MM it reads: “After each incubation time point the colony size were manually measured by a ruler and a picture was taken”.

3) Antimicrobial determination by IMC. Again the work is well done. However for comparison purposes it would be valuable if the authors could provide the same parameters as those used in table 3.

Authors thank Reviewer for her/his suggestion. Unfortunately, we did not evaluate the colony RE: size for each sample treated with different antifungal concentrations as reported in table 3 for the untreated yeasts. Therefore, the same parameter cannot be provided.

4) Line 270-272: it would be worth using the growth rate and calculating doubling time here. This would certainly help readers to picture the difference in replication speed.

RE: Values of growth rate and generation time were added to the discussion paragraph as suggested by the reviewer (line: 271-274).

5) Line 319: I woud remove the statement on combination therapies as it is speculative at this point and no combinations have been used in this study.

RE: The statement on combination therapies has been removed from discussion (line 319) and abstract (line 32), as suggested by Reviewer 2.